# Influence of Cane Molasses Inclusion to Dairy Cow Diets during the Transition Period on Rumen Epithelial Development

**DOI:** 10.3390/ani11051230

**Published:** 2021-04-24

**Authors:** William F. Miller, Evan C. Titgemeyer, Tiruvoor G. Nagaraja, Daniel H. M. Watanabe, Luana D. Felizari, Danilo D. Millen, Zachary K. Smith, Bradley J. Johnson

**Affiliations:** 1Animal Sciences and Industry, Kansas State University, Manhattan, KS 66506, USA; bill@dairycsi.com (W.F.M.); etitgeme@ksu.edu (E.C.T.); tnagaraj@vet.k-state.edu (T.G.N.); 2Department of Animal Production, College of Agricultural and Technological Sciences, São Paulo State University (UNESP), Dracena, São Paulo 17900-000, Brazil; dhw929@mail.usask.ca (D.H.M.W.); luana.doreto@unesp.br (L.D.F.); danilo.millen@unesp.br (D.D.M.); 3Department of Animal Science, South Dakota State University, Brookings, SD 57007, USA; zachary.smith@sdstate.edu; 4Department of Animal and Food Sciences, Texas Tech University, Lubbock, TX 79409, USA

**Keywords:** adaptation, molasses, performance, prepartum, rumen

## Abstract

**Simple Summary:**

The rumen epithelium of dairy cattle undergoes a transformation in response to dietary and physiological changes. Rumen papillae development and adaptation from a typical dry cow diet to a lactating diet can require a substantial amount of time, subsequently limiting the amount of volatile fatty acids (VFA) absorbed from the rumen during early lactation. Infusions of butyrate into the rumen of sheep stimulates cellular proliferation of rumen epithelial tissue, which may prove beneficial for ruminal papillae development in dairy cows during the dry period. However, no studies have investigated mechanisms associated with rumen epithelial adaptation in periparturient dairy cattle. Our hypothesis was that supplementation with cane molasses containing 34% sucrose during the prepartum period would stimulate ruminal butyrate production and ultimately promote ruminal papillae development and absorption rate, thus increasing dry matter intake (DMI) and milk production during the postpartum period. Results from these experiments indicate that diets containing cane molasses during a 60 day dry period can positively influence transition cow performance.

**Abstract:**

The objective of this study was to evaluate the addition of cane molasses during a 60 day dry period on performance and metabolism of Holstein cows during prepartum and postpartum periods. For experiment 1, 26 primiparous and 28 multiparous cows were used. Upon freshening, all cows were offered a common lactation diet. For experiment 2, six multiparous cows fitted with rumen cannulas were used to measure performance and metabolism, following the same protocol as experiment 1. Ruminal propionate increased by 10% during both prepartum and postpartum periods; however, papillae area was greater for cows not fed molasses, and volatile fatty acids (VFA) absorption from the rumen was not increased, resulting in similar glucagon-like-peptide-2 receptor (GLP-2R) density. The improved dry matter intake, when molasses was added into prepartum diets, translated into increased milk yield and energy-corrected milk (ECM) in Experiment 1 only for multiparous cows. For experiment 2, the improvement on milk performance was also observed, where cows fed molasses had 18.5% greater ECM production. Feeding molasses during a 60 day dry period positively influenced transition cow performance, and it was not accompanied by changes in rumen morphometrics; however, this indicates enhanced adaptation by the rumen epithelium based on similar capabilities for VFA absorption.

## 1. Introduction

The rumen epithelium of dairy cattle undergoes a transformation in response to diet change. Rumen papillae development and adaptation from a typical dry cow diet to a lactating diet can require up to 7 weeks postpartum [1] limiting the amount of volatile fatty acids (VFA) absorbed from the rumen during early lactation. It was reported in [2] that surface area of biopsied ruminal papillae increased from 24 mm^2^ at the end of an energy-poor feeding period to over 60 mm^2^ at the end of an energy-rich feeding period; however, when the cows were fed an energy-poor diet again for 5 weeks, the rumen papillae then regressed to 15 mm^2^. Moreover, the same authors observed that the ability of the rumen epithelium to absorb VFA was increased from 4.8 to 16.2 mmol/min when cows were fed an energy-rich diet.

Infusions of butyrate into the rumen of sheep stimulated cellular proliferation in rumen epithelial tissue [3], which might play an important role on ruminal papillae development in dairy cows during the dry period. However, no studies have investigated mechanisms associated with rumen epithelial adaptation in periparturient dairy cattle. Butyrate is an important energy source [4] and an important stimulator of the ruminal epithelium growth [5]. Feeding diets high in starch and sugars, such as provided by molasses, has been used to increase ruminal butyrate to speed up the development of calves’ forestomaches [6]. Furthermore, it is believed that butyrate increases abundance of transcript of proteins mediating VFA absorption [7], and also growth factors secretion [5].

Increases in butyrate levels (9 or 60 mmol/L) supplemented to total parenteral nutrition in neonatal piglets increased plasma glucagon-like-peptide-2 (GLP-2) concentrations and villus height in the jejunum and ileum. Glucagon-like peptide-2 is a 33 AA hormone co-secreted with Glucagon-like peptide-1 (GLP-1) from the enteroendocrine L cells of the small intestine [8]. Cellular actions of GLP-2 are mediated through its receptor (GLP-2R), which is a 7-domain transmembrane protein linked to the G-protein-coupled receptor superfamily. The authors in [8] first reported that GLP-2 had trophic effects in the murine intestine. Studies with GLP-2 have demonstrated increases in mucosal surface area [9,10], reduction in injury [11], restored mucosal integrity, and enhanced intestinal absorptive function [12,13] in rodents. Expression of mRNA for GLP-2R has been detected in tissues of the stomach, small and large intestine, and the central nervous system of monogastric species such as rats, mice, humans, and pigs [14,15], but investigations with ruminant tissues is lacking.

Our hypothesis was that supplementation with cane molasses containing 34% sucrose during the prepartum period would stimulate ruminal butyrate production, ultimately promote ruminal papillae development and valerate absorption rate, increase dry matter intake (DMI) postpartum due to less ruminal VFA accumulation, and thereby improve early lactation performance. In addition, this investigation was conducted to determine the presence and response of GLP-2R in the rumen epithelium during the periparturient period of the dairy cow. Thus, the objective of this study was to evaluate the addition of cane molasses during a 60 day dry period on body weight (BW) changes, blood metabolites, ruminal fermentation pattern, nutrient digestibility, rumen morphometrics, GLP-2R mRNA expression during both prepartum and postpartum periods, and early lactation performance.

## 2. Materials and Methods

### 2.1. Use of Animal Subjects

All procedures were approved by the Kansas State University Institutional Animal Care and Use Committee (Approval # 2430).

### 2.2. Experiment 1

#### 2.2.1. Experimental Design and Treatments

Twenty-six primiparous and 28 multiparous Holstein cows were used to evaluate cane molasses addition to dry cow diets on animal performance. Cows were housed at the Kansas State University Dairy Teaching and Research Center (Manhattan, KS, USA) during the trials. Using a randomized complete block design, cows were blocked by projected calving date and assigned to dietary treatment. Experimental diets were a control diet and a diet with added cane molasses (Table 1). Treatment diets were offered during both the far-off period (day 60 to 30 prior to projected calving) and the close-up period (day 30 to 0 prior to projected calving). During the far-off period, cows were housed in pens, removing opportunity for replication for statistical analysis of DMI as shown on Table 2, and then moved into tie stalls during the close-up period. Cane molasses (3.2% of DM, Table 1) was added to experimental diets after the total mixed ration (TMR) was mixed to add a covering of molasses to all dietary components. Upon freshening, all cows were offered a common lactation diet for the duration of the experiment (through 60 DIM).

#### 2.2.2. Feeding Protocol

Diets were offered twice daily at 0600 and 1600 h as a TMR for ad libitum intake. Amount of TMR offered and refused was recorded daily in order to make the appropriate intake corrections. Dry matter determinations of corn silage and wet corn gluten feed were performed weekly by drying at 105 °C for 24 h, and diets were adjusted to maintain DM proportions of each. Samples of TMR and refusals were collected weekly and dried at 105 °C for 24 h for DM determination. Corn silage, alfalfa hay, whole cottonseed, wet corn gluten feed, and individual grain mixes were sampled weekly and composited by period for analysis of chemical composition by Northeast DHI Forage Testing Laboratory (Ithaca, NY, USA).

#### 2.2.3. Measurement of Body Condition Scored (BCS)

Cows were weighed and body condition scored using a 5-point scale [16] on consecutive days prior to the beginning of each period, immediately following the a.m. milking.

#### 2.2.4. Milking Protocol, Milk Production, and Milk Composition

Cows were milked 4 times daily for the initial 21 DIM then twice daily thereafter with milk yield recorded at each milking. Milk samples were obtained weekly (a.m./p.m. composite) and analyzed for fat, protein, lactose, milk urea nitrogen (MUN), and somatic cell count (SCC) by Heart of America DHI Laboratory (Manhattan, KS, USA). Fat, protein, and lactose content in milk were determined using near infrared spectroscopy (Bentley 2000 Infrared Milk Analyzer, Bentley Instruments Inc., Chaska, MN, USA) as described by [17]. A flow cytometer laser (Somacount 500, Bentley Instruments Inc.) was used to determine SCC [18], and chemical methodology from a modified Berthelot reaction (ChemSpec 150 Analyzer, Bentley Instruments Inc.) was used to measure MUN [19].

Blood Sampling and Analysis. Coccygeal blood was collected into 10 mL EDTA Vacutainer tubes (Becton Dickinson and Co., Franklin Lakes, NJ, USA) at 0900 h (approximately 3 h after feeding) on day 1, 3, 5, 7, and 15 of lactation. Plasma was harvested and stored at −20 °C until future analysis. Plasma samples were analyzed for non-esterified fatty acids (NEFA) content, which was determined with a colorimetric assay (NEFA-C Kit, Wako Chemicals, Richmond, VA, USA) as adapted by [20]; urea nitrogen (PUN), which was determined by a diacetyl-monozime assay using a Technicon Auto Analyzer III (Technicon Industrial Method no. 339-01; Tarrytown, NY, USA) [21]; and glucose concentration, which was determined using a Technicon Auto Analyzer III (Technicon Industrial Method no. SE-4-0036FJ4; Tarrytown, NY, USA) utilizing a peroxidase indicator reaction [22].

Statistical Analysis. Data were analyzed using PROC MIXED of SAS version 8.01 (SAS Institute Inc., Cary, NC, USA). Analyses were completed separately for prepartum data and postpartum data with the model including diet, parity, and diet × parity as fixed effects. Block was included as a random effect. Data on blood composition were analyzed as repeated measures with an autoregressive covariance structure [23], and model included the terms described above as well as sampling day and its interaction with diet, parity, and time × parity. Statistical significance was declared at *p* ≤ 0.05 and trends noted at 0.051 < *p* < 0.10. Least square means were separated using the PDIFF (pairwise comparison) statement in SAS.

### 2.3. Experiment 2

#### 2.3.1. Experimental Design and Treatments

Six multiparous cows fitted with rumen cannulas were used to measure performance, diet digestibility, and rumen parameters. Using a randomized complete block design, cows were blocked by projected calving date and assigned to dietary treatment. The experiment was conducted with second lactation cows that entered a tie-stall barn for adaptation approximately 74 day prior to projected calving and were scheduled to undergo a dry period of 60 day. Cane molasses was added to the far-off TMR at 3.3% of DM and to the close-up TMR at 3.7% of DM. Diets in each period were designed to be isocaloric and isonitrogenous (Table 3).

#### 2.3.2. Feeding Protocol, BCS Evaluation, Milking Protocol, Milk Production, and Milk Composition

The protocols and procedures utilized on the experiment 2 were the same as described for experiment 1.

#### 2.3.3. Blood Sampling and Analysis

Procedures for blood collection were the same as for experiment 1. Plasma was analyzed for NEFA content [20].

#### 2.3.4. Apparent Total Tract Digestion 

Diet digestibility was determined after 14 day on feed in the far-off and close-up periods using acid detergent insoluble ash (ADIA) as an internal marker. Diet and fecal grab samples were obtained over 3 day (day 1: 0200, 0800, 1400, 2000 h; day 2: 0400, 1000, 1600, 2200 h; day 3: 0600, 1200, 1800, 2400 h) and pooled by cow for determination of ADIA content. Diet and fecal samples were dried at 55 °C for 24 h and ground in a Wiley mill to pass a 1 mm screen. Ground samples were then ashed at 450 °C in a muffle oven for determination of ash concentration. Neutral detergent fiber and acid detergent fiber (ADF) contents of diet and feces were determined using the ANKOM filter bag technique (ANKOM Technology Corp., Fairport, NY, USA) [24]. For ADIA determination, ADF samples were ashed at 500 °C in a muffle oven for 8 h and the ADIA concentration was calculated by the residual weight divided by the initial dry sample weight.

#### 2.3.5. Rumen Parameters

Rumen samples were collected 3 h after morning feeding at 60, 30, and 2 day prior to projected calving and 16, 30, 44, 58, and 72 day postpartum. Rumen pH was recorded as an average of three samples obtained from different sites in the rumen using a suction strainer device and portable pH meter. For determination of VFA concentration, 8 mL of rumen fluid and 2 mL of 25% (wt/wt) meta-phosphoric acid were mixed and frozen at −20 °C until analyzed. Ruminal fluid samples were later thawed and centrifuged at 30,000× *g* for 20 min, and a portion of the supernatant fluid was analyzed for VFA concentrations using GC (Hewlett-Packard 5890A, Palo Alto, CA, USA) as described by [25]. 

#### 2.3.6. Valerate Absorption Rate

The valerate absorption rate from the rumen was measured using a technique of valerate dosing [26,27] at 60, 30, and 2 day prior to projected calving and 2, 16, 30, 44, 58, and 72 day postpartum. At 06:30 h, valerate was bolus dosed into the rumen as a solution containing 2.0 mol (204 g) of valeric acid and 4.0 g cobalt (as CoEDTA) adjusted to a pH of 6.0 using sodium hydroxide. Ruminal samples were obtained at 0 h for baseline valerate determination and at 1 and 8 h following the dosing of the valeric acid/Co-EDTA solution. Valerate concentrations were determined as described for VFA above. The rate of valerate disappearance was determined as the negative of the slope of the line when the natural logarithm of valerate concentration was regressed against time. Ruminal fluid Co concentrations were determined by atomic absorption spectrophotometry and were used to calculate liquid passage rates as the negative of the slope of the line when the natural logarithm of Co concentration was regressed against time. Valerate absorption rate was then calculated as: valerate disappearance rate – liquid passage rate. Rumen liquid volume was calculated as the inverse log of the intercept derived from the regression of Co concentration on time.

#### 2.3.7. Rumen Papillae Biopsies

After measurements for determination of valerate absorption rate, on the same days described (except for day 2), rumen papillae morphology was determined. Rumen contents were evacuated through the rumen cannula. A portion of the ventral sac was exteriorized through the rumen fistula and 5 rumen papillae at random within a 2.5 cm × 2.5 cm area were selected and excised. The samples were fixed and embedded in paraffin wax, sectioned (5 μM), and stained with hematoxylin and eosin [28]. The measurements included: papillae height, papillae width, and papillae surface area through the use of a computer-aided light microscope image analysis. Following collection of papillae, a small section of the rumen epithelium from the ventral sac was exteriorized and rinsed with sterile saline to remove debris. Mayo scissors were used to excise a sample of approximately 4.5 cm × 1.5 cm of rumen epithelium without invading the underlying musculature. Biopsy samples were immediately rinsed in a chilled saline solution, snap frozen in liquid N2 and stored at −80 °C for future analysis GLP-2R determination [29]. The area of excision was rinsed with sterile saline and sutured closed using 2-0 vicryl absorbable suture material (Ethicon Inc., Somerville, NJ, USA). The sampling site in the ventral sac area was selected to ensure that the immediate area had not been previously biopsied and that papillae were healthy in appearance. Contents were returned to the rumen upon completion of the biopsies.

#### 2.3.8. Protein Extraction and Western Blot Analysis

Total protein extraction from tissue was completed in the lab by homogenizing ~500 mg of sample for 30 s in 2.5 mL M-PER containing 8.8 (wt/vol) g/L NaCl (Pierce Biotechnology, Rockford, IL). Homogenate was then centrifuged at 10,000× *g* for 15 min to pellet debris. Aliquots of supernatant were transferred to 2 tubes, flash frozen in liquid N2, and stored at −80 °C until analysis. Concentration of protein was later determined at a wavelength of 260/280 nm using a ND-1000 Spectrophotometer (Nano-Drop Technologies, Wilmington, DE, USA). Sixty micrograms of total protein was separated by gel electrophoresis using SDS-PAGE with precast Pierce Precise Protein 4–20% Gradient Gels (Pierce Biotechnology, Rockford, IL, USA), and separated proteins were transferred onto a nitrocellulose membrane using a Trans-Blot Semi Dry Electrophoretic Transfer system (Bio-Rad, Hercules, CA, USA). Nitrocellulose membranes were blocked using a blocking buffer (Starting Block Buffer, Pierce Biotechnology, Rockford, IL, USA) for 15 min at 37 °C. Polyclonal rabbit primary antibody against GLP-2R (LS-A1312; Lifespan Biosciences, Seattle, WA, USA) in 5 mL of blocking buffer was added to the membrane and incubated overnight at 4 °C. The nitrocellulose membrane was then washed 3× for 15 min each with phosphate buffered saline-Tween 20. Secondary antibody, goat anti-rabbit-HRP (sc-2004; Santa Cruz Biotechnology, Inc., Santa Cruz, CA, USA), was added in 1 mL of blocking buffer to the membrane and incubated for 1 h. Detection of GLP-2R was completed using chemiluminescence with a Fluorchem 8800 Imaging System (AlphaInnotech, San Leandro, CA, USA).

#### 2.3.9. Sample Preparation, RNA Isolation, and Real-time Quantitative-PCR

Total RNA was isolated from 100 mg of tissue samples using sterile steel mortar bowls cooled by liquid N2. Tissue samples were homogenized using a sterile pestle in liquid N2. Then, 3 mL TRI Reagent (Sigma, St. Louis, MO, USA) was added to the ground tissue sample, and 1 mL of tissue in TRI Reagent was incubated at room temperature for 5 min. Following incubation, chloroform (Sigma, St. Louis, MO, USA) was added and samples were centrifuged for 15 min at 12,000× *g* at room temperature. Following centrifugation, the top layer was removed and transferred to a new microcentrifuge tube. Isopropanol (Sigma, St. Louis, MO, USA) was added, and samples were centrifuged for 10 min at 12,000× *g* to isolate the RNA pellet. The RNA pellet was then treated to remove any contaminating genomic DNA using the DNA-free kit (Ambion, Austin, TX, USA). The RNA concentration was determined by absorbance at 260 nm and integrity of RNA was determined by gel electrophoresis. Total RNA with ethidium bromide was loaded onto a 1% agarose gel to separate and visualize 28S and 18S rRNA bands. Real-time quantitative-PCR was used to measure the quantity of GLP2-R gene expression relative to the quantity of 18S ribosomal RNA (rRNA) in total RNA isolated from tissue. Measurement of the relative quantity of cDNA was performed using TaqMan Universal PCR Master Mix (Applied Biosystems, Foster City, CA, USA), 900 nM of the appropriate forward and reverse primers, 200 nM of appropriate TaqMan detection probe, and 1 µL of the cDNA mixture. The bovine specific GLP2-R forward (GTGAGACAGAGTGGCTGTCCTATG) and reverse (TGCCCACAAAGTAGTGCAAGC) primers and TaqMan (6FAM-TTGCTGCCTCCTGCCGCTCA-TAMRA) detection probes were synthesized using published GenBank sequences. Commercially available eukaryotic 18S rRNA primers and probes were used as an endogenous control (Applied Biosystems; Genbank Accession #X03205). The ABI Prism 7000 detection system (Applied Biosystems, Foster City, CA, USA) was used to perform the assay utilizing the thermal cycling variables recommended by the manufacturer (50 cycles of 15 s at 95 °C and 1 min at 60 °C). The 18S rRNA endogenous control was used to normalize the expression of GLP2-R [29].

#### 2.3.10. Statistical Analysis

Analyses were completed separately for prepartum data and postpartum data. Data were analyzed using PROC MIXED of SAS version 8.01 (SAS Institute Inc., Cary, NC, USA). Responses without repeated observations were analyzed using a model containing the fixed effect of diet, and block was considered as a random effect. Responses with measurements over time were analyzed as repeated measures with an autoregressive covariance structure [23], and the model included the terms described above as well as sampling day and its interaction with diet. Digestibility and prepartum rumen fluid measurements were analyzed with the model including diet, period (far-off vs. close-up), and diet × period. Statistical significance was declared at *p* ≤ 0.05 and trends noted at 0.05 < *p* < 0.10. Least square mean separation for treatment and parity effects were conducted using the PDIFF statement in SAS, all means are presented as the least square mean and the corresponding standard error of the mean (SEM).

## 3. Results

### 3.1. Experiment 1

#### 3.1.1. Dry Matter Intake, Milk Yield, and Milk Composition

Intake of DM was greater (*p* < 0.0001) for multiparous cows than for primiparous cows (Table 2). Moreover, the addition of cane molasses to the dry cow diets increased DMI (*p* < 0.0001) during the close-up period, and this translated to an increase (*p* < 0.0001) in DMI during the postpartum period. Significant diet × parity interactions were observed for milk yield (*p* = 0.003), energy-corrected milk (ECM) (*p* = 0.003), ECM/DMI (*p* = 0.001), and milk fat yield (*p* = 0.02; Table 2). Increases (*p* < 0.0001) in milk yield, milk fat yield, and ECM were observed for multiparous cows previously fed the molasses diet during the dry period, but not for primiparous cows. Multiparous cows previously fed molasses improved ECM/DMI; however, the addition of molasses into prepartum diets impaired ECM/DMI of primiparous cows. Percentage fat in milk was greater for primiparous cows than for multiparous cows (*p* = 0.01), but no differences were noted for percentage protein in milk (*p* = 0.88). Yield of protein from milk was greater for multiparous cows than for primiparous cows. Milk fat and protein percentages were not affected by prepartum diet. Feeding molasses during the dry period increased the yield of protein (*p* = 0.0006) in milk and concentration of MUN was greater (*p* = 0.05) for cows previously fed molasses, as shown in Table 2. No main effects of diet (*p* = 0.99) and parity (*p* < 0.14) were observed for percentage lactose in milk; however, multiparous cows had greater concentration of MUN (*p* < 0.0001) and showed a tendency to decrease SCC (*p* = 0.09).

#### 3.1.2. Body Weight and Body Condition

As expected, BW was greater (*p* ≤ 0.004) for multiparous cows during the study; however, BCS was greater (*p* < 0.001) for primiparous cows throughout the experiment (Table 4). A tendency (*p* = 0.08) was observed on day 30 for cows fed with molasses to have higher BW than cows fed with the control diet. Significant changes in body condition were observed between primiparous and multiparous cows for the intervals from day −60 to −30 (*p* = 0.03), 0 to 30 (*p* = 0.001), and 0 to 75 (*p* = 0.04), where primiparous cows lost more body condition than multiparous ones (Table 5).

#### 3.1.3. Plasma Metabolites

Effects of experimental diets on postpartum plasma constituents are shown in Table 6. No interactions (*p* > 0.10) between collection day and parity or diet were observed for any of the variables evaluated. Plasma urea-N concentration was greater (*p* < 0.0001) for multiparous cows than for primiparous cows. Plasma glucose concentration tended (*p* = 0.09) to be greater and NEFA were greater (*p* = 0.002) for primiparous cows than for multiparous cows.

### 3.2. Experiment 2

#### 3.2.1. Dry Matter Intake, Milk Yield, and Milk Composition

DMI, milk yield, and milk composition are reported in Table 7. Molasses did not improve DMI when added to the far-off diet, but DMI during the close-up period was greater (*p* = 0.002) for cows fed molasses diets than for cows fed control diets. Likewise, a tendency (*p* = 0.08) was observed for DMI postpartum, in which cows fed molasses during the prepartum period had greater DMI than cows receiving the control diet. The improvement in DMI for cows fed the molasses diet did not translate into a significant improvement in milk yield in this experiment (*p* = 0.41), although yield of ECM tended to be greater (*p* = 0.07) for cows fed molasses than for cows fed control diets during the prepartum period. Percentage of fat in milk did not differ between treatments, whereas percentage of protein in milk tended to be greater (*p* = 0.06) for cows fed molasses. Yield of fat was greater (*p* = 0.002) and yield of protein tended to be greater (*p* = 0.07) for cows fed molasses compared to cows fed control diets. In addition, MUN concentration was greater (*p* = 0.001) for cows consuming the molasses diet during the prepartum period. On the other hand, addition of molasses into prepartum diets tended to reduce the percentage of lactose in milk (*p* = 0.06). No diet effect was detected for SCC (*p* = 0.22).

#### 3.2.2. Body Weight and Body Condition

Initial cow BW and BCS are reported in Table 8 and Table 9. The BW and BCS were not different between treatments, and changes in BW and BCS over time did not significantly differ between cows fed control or molasses diets.

#### 3.2.3. Plasma Metabolites

No effect of diet molasses was observed for NEFA concentration during the first 15 DIM (*p* = 0.25). A day effect was observed (*p* = 0.04), where NEFA concentrations decreased between day 3 and 15 DIM (Figure 1). No interaction between diet and days of blood collection was noted for plasma NEFA concentrations (Figure 1). 

#### 3.2.4. Diet Digestibility

Digestibility of organic matter (OM) was greater (*p* = 0.05) for diets offered during the close-up period relative to those offered during the far-off period (Table 10). The addition of molasses to dry cow diets did not affect digestibility of OM for this experiment. Digestibilities of neutral detergent fiber (NDF) and acid detergent fiber (ADF) did not differ between the far-off and the close-up periods nor were they affected by molasses addition. 

#### 3.2.5. Rumen Fluid VFA Proportions and Kinetics

Rumen fluid prepartum measurements and kinetics are presented in Table 11. No diet main effect was observed (*p* > 0.10) for any of rumen parameters evaluated in this study. During the close-up period, rumen pH was lower (*p* = 0.04) than during the far-off period. Total VFA concentration was greater (*p* = 0.02) in rumen fluid from cows during the close-up period compared to rumen fluid from cows during the far-off period. Ruminal molar acetate percentage was greater (*p* = 0.04) during the far-off period compared to the close-up period; however, molar propionate percentage in ruminal fluid was greater (*p* = 0.003) during the close-up period relative to the far-off period. Percentage of butyrate and isovalerate, as well as liquid dilution, rumen liquid volume, liquid outflow, and liquid turnover did not differ between the far-off and close-up periods. Valerate absorption was greater (*p* = 0.02) during the close-up period compared to the far-off period. 

Postpartum rumen measurements are shown in Table 12. As for prepartum measurements, no main effect of diet was observed (*p* > 0.10) for any of the rumen parameter evaluated in this study. Postpartum ruminal propionate demonstrated change over time (*p* = 0.02), with the molar percentage being lowest on day 2 and greatest on day 16. Day was also significant for liquid dilution rate (*p* = 0.03) and liquid outflow (*p* = 0.002), which were both increased between day 2 and 16, and then decreased from day 44 to 72.

#### 3.2.6. Rumen Papillae Morphology

Prepartum rumen papillae measurements are shown in Figure 2, Figure 3 and Figure 4. Rumen papillae length was longer (*p* < 0.001) for cows fed the control diet compared to cows fed molasses during the prepartum period (Figure 2). Width of rumen papillae was unaffected (*p* = 0.63) by prepartum diet; however, papillae width was greater (*p* = 0.02) for the close-up period relative to the far-off period (Figure 3). Feeding molasses during the dry period did not enhance rumen papillae surface area as cows fed the control diet had greater (*p* = 0.02) papillae surface area compared to cows fed the molasses diet (Figure 4).

Postpartum rumen papillae measurements are shown in Figure 5, Figure 6 and Figure 7. A significant interaction between diet and day was observed for papillae length (*p* = 0.009), width (*p* = 0.0003), and surface area (*p* < 0.0001), where papillae from cows that were previously fed control were longer, wider, and contained greater surface area compared to the papillae from cows previously fed molasses on day 30, 44, 58, and 72.

#### 3.2.7. GLP-2R Western Blot

Primary antibody was polyclonal rabbit antibody raised against human antigen; because of the conserved nature of GLP-2R, its use is likely acceptable. A query using BLAST in Pub Med returned a homology of >93% for human and bovine GLP-2R. Representative Western blots of prepartum and postpartum GLP-2R are shown in panel A of Figure 8 and Figure 9, respectively. The glycosylated form of GLP-2R is 76 kDa in size, which matches the size of proteins identified as GLP-2R in the blots. Prepartum GLP-2R Western blot density was unaffected by diet or day (Figure 8, Panel B). Blot density for GLP-2R during lactation is represented in Figure 9. Cows previously offered the control diet during the prepartum period had greater (*p* < 0.01) receptor density during the postpartum period (Figure 9, Panel B).

#### 3.2.8. mRNA Expression

A diet × day interaction (*p* = 0.05) during the prepartum period was observed for expression of mRNA for GLP-2R (Figure 10). Initial rumen epithelial tissue samples taken on day −60 before offering experimental diets were similar for expression of mRNA (Figure 10). After the initial 30 day of feeding control or molasses diets and immediately prior to the close-up period, mRNA expression was elevated for cows fed the molasses diet (*p* < 0.001). Messenger RNA expression encoding GLP-2R declined as parturition approached on day –2 (Figure 10). Postpartum expression of mRNA for GLP-2R is represented in Figure 11. The main effect of day tended (*p* = 0.08) to differ during the postpartum period for expression of mRNA encoding GLP-2R, whereas no effect of diet was observed (*p* = 0.13).

## 4. Discussion

The two experiments were conducted to determine if the prepartum dietary addition of cane molasses, a source of sucrose, could influence rumen epithelial adaptation and subsequently lactation performance. The hypothesis was that an increase in ruminal butyrate would positively influence rumen papillae development, subsequently enhancing VFA absorption and easing the transition to the lactation diet containing a greater proportion of rapidly fermented carbohydrates, resulting on improved lactation performance. Absorption of VFA from the rumen pool is commonly attributed to the development of papillae lining the inside of the rumen. Greater surface area leads to greater absorption [2] and attenuates VFA accumulation, which can cause ruminal acidosis. However, the addition of molasses into prepartum diets did not cause any significant changes in molar butyrate percentage during either prepartum or postpartum periods (Table 11 and Table 12). In addition, papillae length, and consequently papillae surface area, was greater for cows not fed molasses (Figure 2, Figure 4, Figure 5 and Figure 7). Diets utilized in Experiment 2 were designed to be isocaloric and DMI was greater, at least during the close-up period, for the molasses-containing diet; thus, a response to energy intake per se was not expected. However, papillae response was consistent with rumen VFA concentrations in that total VFA concentration was numerically less for the molasses-containing diet prepartum, demonstrating a positive relationship between VFA concentrations and rumen papillae length and surface area. As a result, VFA absorption from the rumen was not increased during both the prepartum or postpartum periods (Table 11 and Table 12) in this study. Dietary strategies that promote the development of the rumen papillae would be advantageous to the animal by improving the rumen’s ability to adapt to diet changes, specifically regarding the diet switch from late gestation to lactation. In the current study, despite not significant, molar butyrate percentage increased by 5% during the prepartum period, and molar propionate percentage significantly increased by 10% during both periods. According to [30], VFA is proportionately responsible for promoting the metabolically active growth of ruminal papillae, especially propionate. Logically, it would be appropriate to employ a dietary strategy to promote rumen epithelial development, prior to parturition and for a sufficient duration to allow for rumen papillae transition to take place, which may not have occurred in this study.

The increases in prepartum close-up DMI in response to molasses (Table 3 and Table 7) are consistent with improvements in rumen adaptation during the prepartum period for the upcoming lactation diet that contains greater proportions of concentrates. Increases in prepartum DMI have been observed when increasing dietary concentrate levels in numerous studies [31,32,33]. Likewise, sucrose addition to the lactation diet has stimulated DMI during the early lactation period [34]. This increase in DMI should stimulate the secretion of gut peptides from the small intestine that are responsible for metabolic and physiological adaptations to aid nutrient absorption. Actions of these circulating gastrointestinal hormones are mediated through the expression of specific receptors such as GLP-2R. The increase in DMI prepartum when cows were fed molasses was associated with increased GLP-2R mRNA expression in rumen epithelium tissue on day −30 (Figure 10); however, it did not positively affect the GLP-2R density (Figure 8), which explains why rumen epithelium did not respond positively to the addition of molasses into prepartum diets. Moreover, with the decrease of molasses inclusion from approximately 3.5% (close-up diets) to 0.93% (lactation diets; Table 1 and Table 2), the GLP-2R density (Figure 9) decreased on molasses-fed cows during the postpartum period, which resulted in smaller papillae area surface (Figure 7) for VFA absorption. Diminishing expression of mRNA for GLP-2R (Figure 11) over the initial 72 day of lactation may reflect an adaptation in gut epithelium to intake and nutrient absorption. The initial increase in mRNA during early lactation for cows offered the control diet prepartum may suggest a delayed response by gut tissue of these cows as mRNA expression peaked at day 30 prepartum for cows fed molasses. Although the presence of GLP-2R in the rumen suggests that GLP-2 may play an important role in rumen development, the changes in GLP-2R over time or in response to dietary treatment are unlikely to tell the complete story. Clearly, the concentrations of GLP-2 also need to be considered. It is possible that GLP-2R is regulated either directly or indirectly by changes in blood concentrations of GLP-2. Moreover, the tissue responsiveness to GLP-2 will be dependent upon the concentrations of the receptor and of the agonist. [35] reported that mRNA expression of target genes related to VFA absorption was regulated by consequences of low ruminal pH, which was not the case of our study.

In feedlot cattle, when rumen epithelium samples are collected from the cranial sac (greater papillae density when compared to ventral sac), the number of papillae per square centimeter decreases, papillae surface area, as well as rumen wall absorptive surface area, increases when a higher amount of rapidly fermentable carbohydrates are added to diets [36,37]. Although the number of papillae per square centimeter of rumen wall was not measured in this study, the rumen wall absorptive surface area, which is the rumen morphometric variable most correlated with VFA absorption, is closely associated with papillae surface area, which may explain why valerate absorption rate from rumen to the bloodstream did not increase in this study when molasses was added into the diets. On the other hand, although rumen papillae from cows previously fed molasses-containing diets were smaller than those fed the control diet, valerate absorption from the rumen was not affected. Thus, it is possible that molasses-containing diets may have increased papillary density, which may have offset the effects of papillary surface area. Rumen papillae adaptation during the prepartum period appears to influence valerate absorption, but the response does not appear to be proportional. The relatively low inclusion of starch in prepartum diets, and the high inclusion of roughage, may not allow adequate production of propionate and butyrate in the rumen to positively affect rumen epithelium development. The absorptive surface area of ruminal epithelium may become larger when absorption capacity is exceeded, so feeding diets containing higher amounts of energy or including greater amounts of molasses in the prepartum diets may lead to more consistent ruminal epithelium response.

In our study, adding molasses to prepartum diets resulted in greater prepartum DMI, which translated to greater postpartum DMI. Because decreasing the DMI depression immediately prepartum is positively correlated with reduced circulating NEFA concentrations and postpartum DMI [38], it was expected that molasses-containing diets fed during the dry period would decrease circulating concentrations of NEFA in this study. However, the addition of molasses to prepartum diets did not reduce NEFA concentration in blood (Table 6, Figure 1). When molasses was added to prepartum diets, the increased DMI during early lactation translated into increased yield of milk and ECM in Experiment 1 for multiparous, but not for primiparous cows, (Table 2). The improvement in milk production was also observed on Experiment 2, where cows fed molasses increased ECM by 18.5% (Table 7), although replication was not adequate in Experiment 2 to confidently assess lactational performance. Yields of milk fat and milk protein were directly related to milk production as differences between treatments in milk composition were not observed. Apparently, molasses inclusion was effective in improving lactation performance for multiparous cows, although the inclusion level of approximately 3.5% of dietary DM evaluated in this study was not sufficient to alter rumen papillae development. The fact that primiparous cows did not improve lactational performance in response to addition of molasses to prepartum diets may be related to the body tissue mobilization (Table 5), which contributed to the higher levels of circulating NEFA in those cows (Table 6). Primiparous cows produce significantly less milk than multiparous cows (Table 2) yet can mobilize more body tissue (Table 5), which makes their lactational performance in early lactation more resistant to changes in feed intake than that of multiparous cows. 

## 5. Conclusions

Results from these experiments indicate that diets containing cane molasses during a 60 day dry period can positively influence transition cow performance. These studies indicate that increasing DMI during the prepartum and postpartum periods is beneficial to an efficient transition to lactation and that the addition of cane molasses may stimulate intake during this time. Increased postpartum DMI for cows previously offered a molasses diet was not accompanied by an increase in ruminal butyrate concentrations or valerate absorption rate from the rumen; however, there may have been some enhanced adaptations by the rumen epithelium, because rumen papillae from cows previously fed molasses diets were smaller. Therefore, the paradigm of greater rumen papillae surface area being fundamental to VFA absorption from the rumen was not supported by our data. The lack of a positive relationship between rumen papillae surface area and valerate absorption warrants further investigation, as well as the role of GLP-2R in rumen epithelial development. 

## Figures and Tables

**Figure 1 animals-11-01230-f001:**
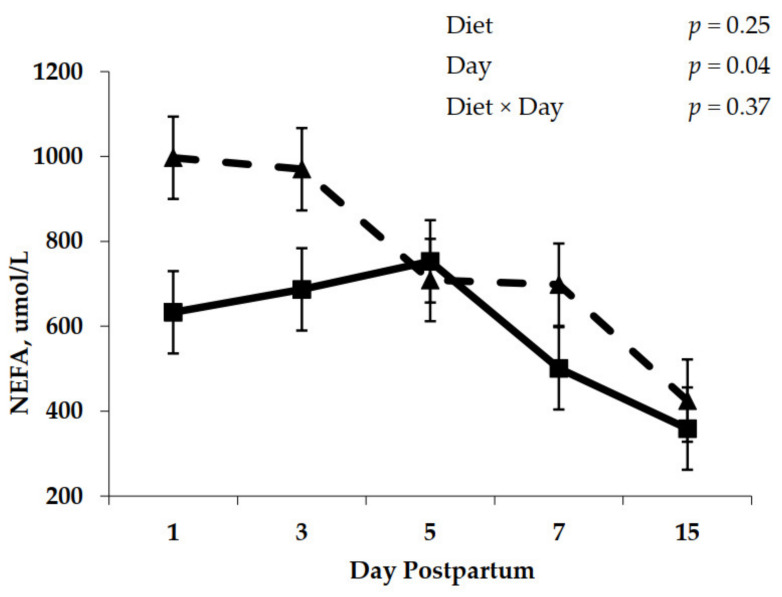
Postpartum plasma non-esterified fatty acids (NEFA) concentration for cows fed control (triangles) or molasses-containing (squares) diets during a 60 day dry period (error bars = pooled standard error of the mean; Experiment 2, *n* = 3)

**Figure 2 animals-11-01230-f002:**
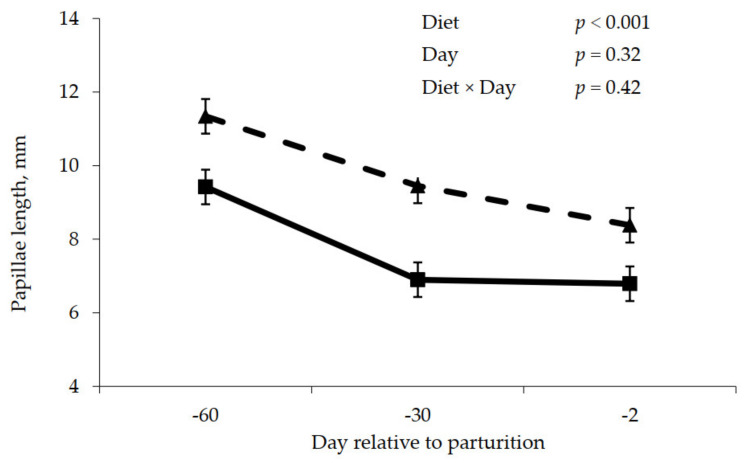
Prepartum papillae length for cows fed control (triangles) or molasses-containing (squares) diets during a 60 day dry period (error bars = pooled standard error of the mean, Experiment 2, *n* = 3).

**Figure 3 animals-11-01230-f003:**
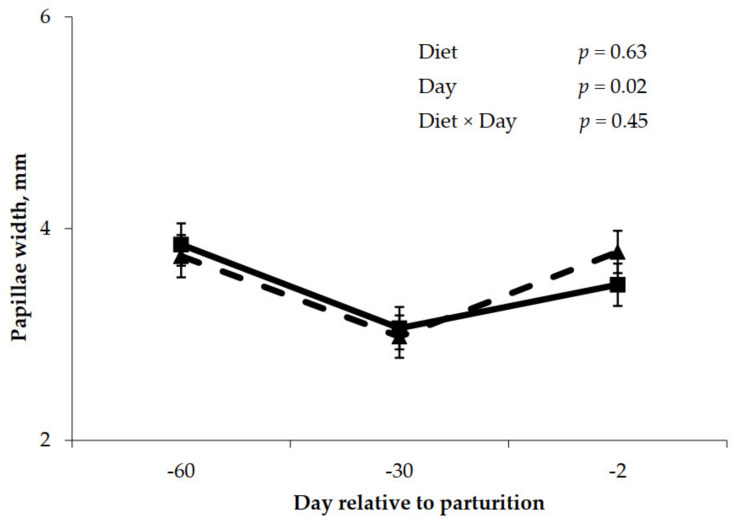
Prepartum papillae width for cows fed control (triangles) or molasses-containing (squares) diets during a 60 day dry period (error bars = pooled standard error of the mean, Experiment 2, *n* = 3).

**Figure 4 animals-11-01230-f004:**
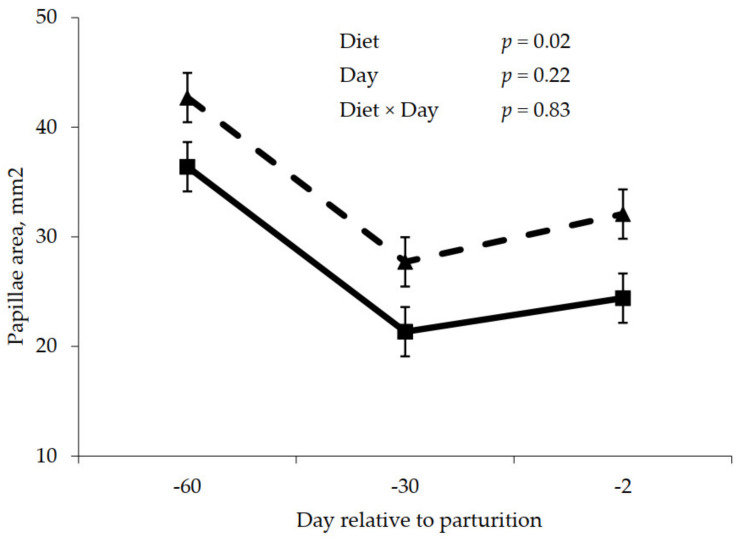
Prepartum papillae surface area for cows fed control (triangles) or molasses-containing (squares) diets during a 60 day dry period (error bars = pooled standard error of the mean, Experiment 2, *n* = 3).

**Figure 5 animals-11-01230-f005:**
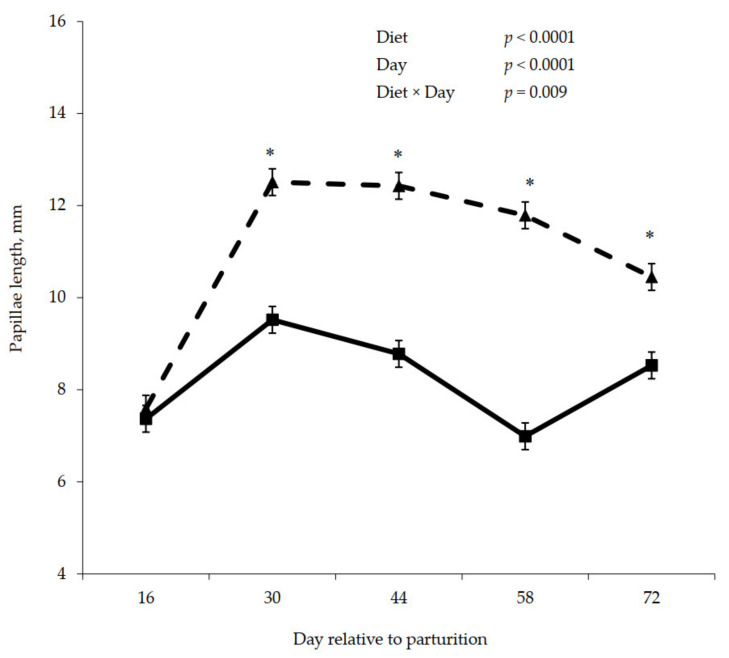
Postpartum rumen papillae length for cows fed control (triangles) or molasses-containing (squares) diets during a 60 day dry period (error bars = pooled standard error of the mean, Experiment 2, *n* = 3). * Within each day, means differ at *p* ≤ 0.05.

**Figure 6 animals-11-01230-f006:**
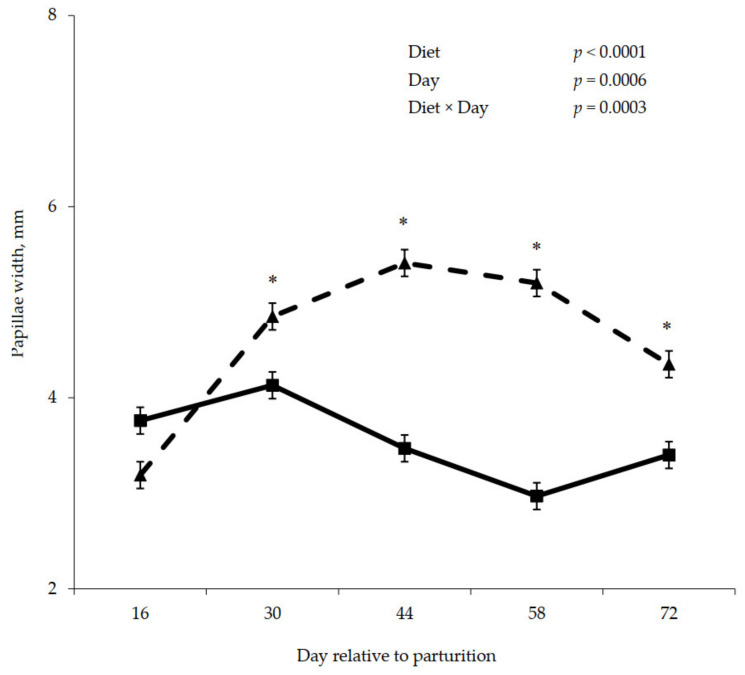
Postpartum rumen papillae width for cows fed control (triangles) or molasses-containing (squares) diets during a 60 day dry period (error bars = pooled standard error of the mean, Experiment 2, *n* = 3). * Within each day, means differ at *p* ≤ 0.05.

**Figure 7 animals-11-01230-f007:**
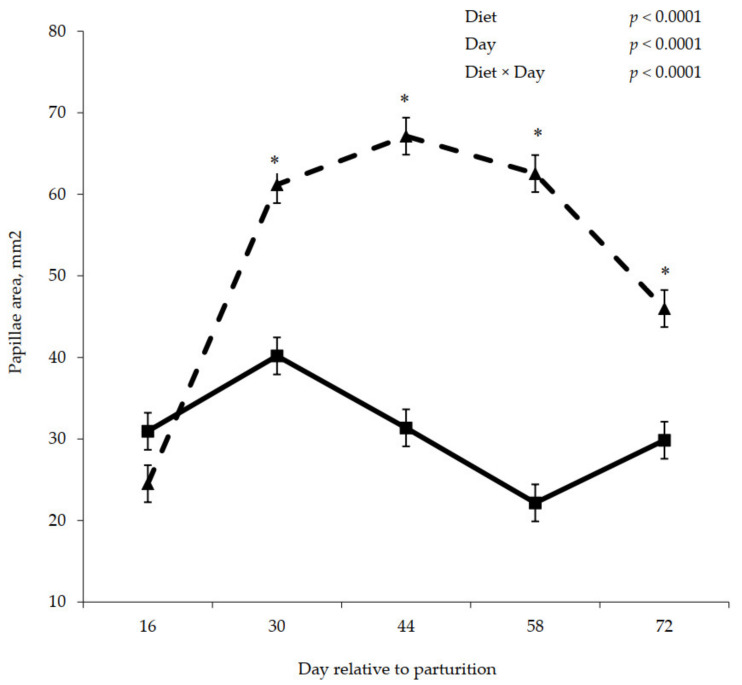
Postpartum rumen papillae surface area for cows fed control (triangles) or molasses-containing (squares) diets during a 60 day dry period (error bars = pooled standard error of the mean, Experiment 2, *n* = 3). * Within each day, means differ at *p* ≤ 0.05.

**Figure 8 animals-11-01230-f008:**
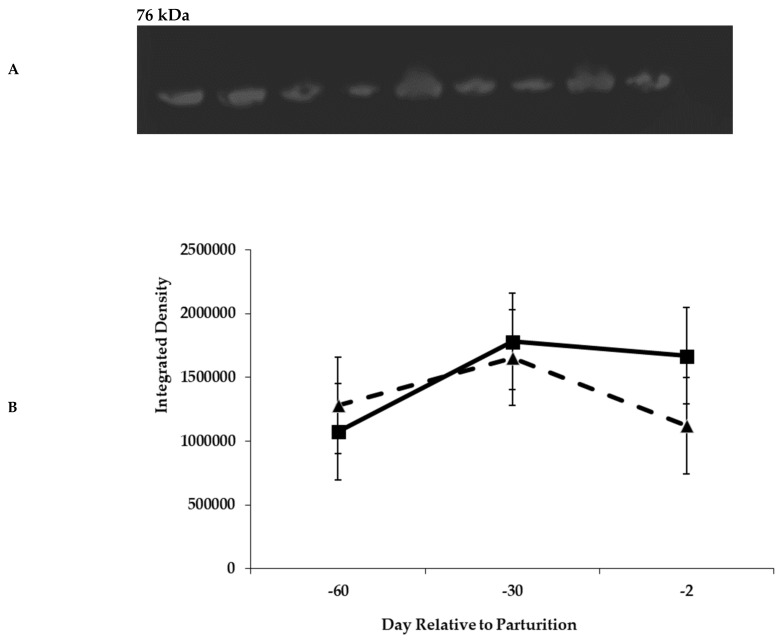
Representative Western blot analysis of glucagon-like peptide-2 receptor (GLP2R) from rumen epithelium during the prepartum period (Panel **A**, Appendix A). Glucagon-like peptide-2 receptor density in rumen epithelial tissue from cows fed control (triangles) or molasses-containing (boxes) diets during the prepartum period (Diet, *p* = 0.60; Day, *p* = 0.39; Diet × Day, *p* = 0.62; error bars = pooled standard error of the mean, Experiment 2, *n* = 3; Panel **B**).

**Figure 9 animals-11-01230-f009:**
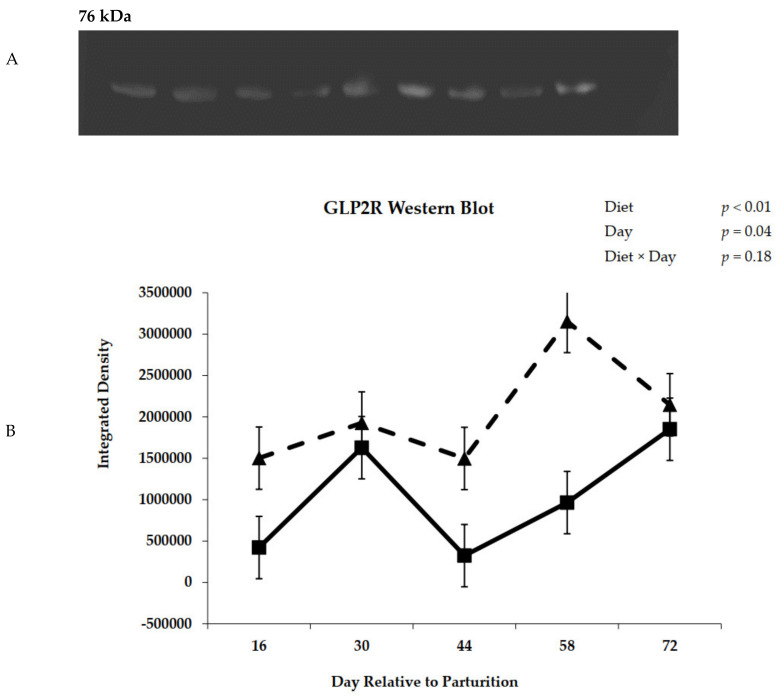
Representative Western blot analysis of glucagon-like peptide-2 receptor from rumen epithelium during postpartum period (Panel **A**, Appendix A). Glucagon-like peptide-2 receptor density in rumen epithelial tissue from cows fed control (triangles) or molasses-containing (boxes) diets during the postpartum period (Diet, *p* < 0.01; Day, *p* = 0.04; Diet × Day, *p* = 0.18; error bars = pooled standard error of the mean, Experiment 2, *n* = 3; Panel **B**).

**Figure 10 animals-11-01230-f010:**
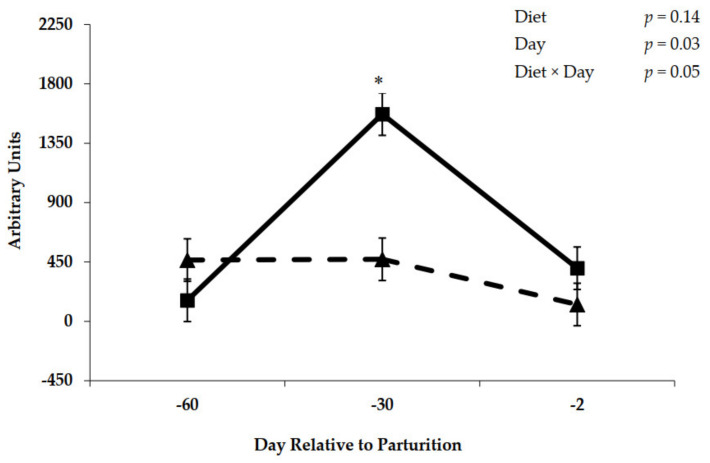
Glucagon-like peptide-2 receptor mRNA expression in rumen epithelial tissue from cows fed control (triangles) or molasses-containing (boxes) diets during the prepartum period. (Error bars = pooled standard error of the mean, Experiment 2, *n* = 3.) * Within each day, means differ at *p* < 0.001.

**Figure 11 animals-11-01230-f011:**
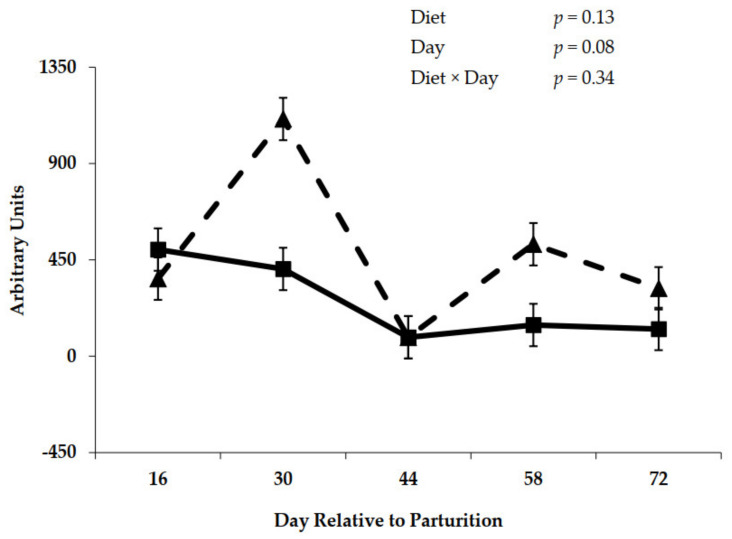
Glucagon-like peptide-2 receptor mRNA expression in rumen epithelial tissue from cows fed control (triangles) or molasses-containing (boxes) diets during the postpartum period. (Error bars = pooled standard error of the mean, Experiment 2, *n* = 3.).

**Table 1 animals-11-01230-t001:** Ingredient and nutrient composition of control and cane molasses-containing diets offered during a 60 day dry period for Experiment 1.

	Far-off ^1^		Close-up ^1^		Lactation
	Control	Molasses	Control	Molasses	
Ingredient	% DM ^2^				
Prairie hay	56.3	56.3	36.8	36.8	-
Corn silage	13.0	13.0	13.1	13.1	22.0
Wet corn gluten feed	9.7	9.7	12.0	12.0	20.4
Soybean meal, solvent	7.0	7.0	5.8	5.8	-
Soybean meal, expeller	-	-	-	-	7.6
Alfalfa hay	6.7	6.7	9.0	9.0	15.3
Whole cottonseed	3.2	3.2	5.0	5.0	8.5
Corn, ground	3.2	-	16.4	13.1	20.4
Molasses, sugarcane	-	3.2	-	3.3	0.95
Sodium bicarbonate	-	-	0.48	0.47	0.80
Magnesium oxide	0.12	0.12	0.40	0.40	0.20
Fishmeal, menhaden	-	-	1.20	1.20	2.12
Limestone	0.54	0.53	0.14	0.14	1.40
Trace mineralized salt ^3^	0.40	0.40	0.20	0.20	0.34
Vitamin premix ^4^	0.14	0.14	0.14	0.14	0.12
Sodium selenite ^5^	0.030	0.030	0.020	0.020	0.008
Nutrient					
DM, % as fed	72.6	71.7	70.3	68.9	61.6
Crude Protein (CP), %	12.3	12.1	14.1	14.5	19.1
Neutral detergent fiber (NDF), %	53.7	52.5	43.9	43.2	30.6
Acid detergent fiber (ADF), %	33.3	32.5	26.8	25.8	17.1
NE_L_, Mcal/kg ^6^	1.47	1.47	1.64	1.63	1.69
Calcium, %	0.61	0.62	0.54	0.53	1.00
Phosphorus, %	0.36	0.42	0.44	0.51	0.51
Magnesium, %	0.34	0.32	0.45	0.42	0.41
Potassium, %	1.56	1.53	1.41	1.39	1.21
Sodium, %	0.27	0.26	0.25	0.22	0.43

^1^ Far-off period (day 60 to 30 prior to projected calving) and close-up period (day 30 to 0 prior to projected calving); ^2^ Dry matter; ^3^ Composition not less than: 95.5% NaCl, 0.24% Mn, 0.24% Fe, 0.05% Mg, 0.032% Cu, 0.032% Zn, 0.007% I, and 0.004% Co; ^4^ Provided 4400 IU Vitamin A; 2200 IU Vitamin D; and 16 IU Vitamin E per kg of dietary DM; ^5^ Provided 0.06 mg Se per kg of diet DM; ^6^ Net energy for lactation.

**Table 2 animals-11-01230-t002:** Dry matter intake (DMI) and milk yield and composition during the initial 60 days of lactation for primiparous and multiparous cows fed control or molasses-containing diets during a 60 day dry period (Experiment 1).

	Control		Molasses		SEM	*p*Value		
	Primiparous	Multiparous	Primiparous	Multiparous		Diet	Parity	Diet × Parity
*n*	13	14	13	14				
DMI, kg/day								
Far-off	11.8	14.9	11.9	15.6	-	-	-	-
Close-up	11.6	16.7	12.9	18.5	0.32	<0.0001	<0.0001	0.45
Lactation	16.9	24.3	18.3	25.7	0.25	<0.0001	<0.0001	0.89
Milk yield, kg/day	35.8	46.4	36.7	50.8	0.57	<0.0001	<0.0001	0.003
ECM, kg/day ^1^	36.3	45.1	37.2	49.5	0.58	<0.0001	<0.0001	0.003
ECM/DMI	2.28	1.94	2.14	2.01	0.029	0.27	<0.0001	0.001
Milk fat								
%	3.84	3.43	3.78	3.51	0.135	0.93	0.01	0.60
kg/day	1.33	1.56	1.36	1.75	0.033	0.003	<0.0001	0.02
Milk protein								
%	2.89	2.96	2.90	2.86	0.078	0.54	0.88	0.45
kg/day	1.02	1.35	1.06	1.43	0.015	0.0006	<0.0001	0.13
Milk lactose, %	4.81	4.78	4.84	4.75	0.039	0.99	0.14	0.45
SCC, per µL ^2^	266	125	272	264	49.1	0.10	0.09	0.14
MUN, mg/Dl ^3^	14.3	16.3	14.7	16.8	0.26	0.05	<0.0001	0.78

^1^ Energy corrected milk; ^2^ Somatic cell count; ^3^ Milk urea nitrogen. SEM: Standard error of the mean.

**Table 3 animals-11-01230-t003:** Ingredient and nutrient composition of control and cane molasses-containing diets offered during a 60 day dry period for Experiment 2.

	Far-off		Close-up		Lactation
	Control	Molasses	Control	Molasses	
Ingredient	------------------------------ % DM ------------------------------				
Prairie hay	57.5	56.6	37.3	37.3	-
Corn silage	12.6	12.4	12.2	12.2	20.5
Wet corn gluten feed	9.8	9.6	11.9	11.9	20.6
Corn, ground	1.3	-	16.4	12.9	21.7
Alfalfa hay	6.8	6.7	9.3	9.3	15.5
Whole cottonseed	3.4	3.0	5.1	5.1	8.4
Soybean meal, solvent	7.1	7.0	5.1	5.1	-
Soybean meal, expeller	-	-	-	-	7.5
Molasses, sugarcane	-	3.3	-	3.7	0.90
Sodium bicarbonate	-	-	0.48	0.46	0.80
Magnesium oxide	0.14	0.13	0.40	0.40	0.20
Fishmeal, menhaden	-	-	1.25	1.23	2.08
Limestone	0.54	0.53	0.14	0.14	1.36
Trace mineralized salt ^1^	0.53	0.52	0.20	0.19	0.33
Vitamin premix ^2^	0.14	0.14	0.17	0.16	0.12
Sodium selenite ^3^	0.030	0.030	0.020	0.020	0.008
Nutrient					
DM ^4^, % as fed	73.2	73.3	70.9	71.6	62.5
Crude Protein (CP), %	12.4	12.2	14.0	13.9	18.8
Neutral detergent fiber (NDF), %	53.7	52.4	46.9	45.8	35.8
Acid detergent fiber (ADF), %	33.4	32.6	26.5	25.9	19.1
NE_L_, Mcal/kg	1.25	1.29	1.41	1.42	1.63
Calcium	0.51	0.52	0.43	0.46	0.94
Phosphorus	0.34	0.35	0.39	0.39	0.51
Magnesium	0.39	0.29	0.44	0.45	0.42
Potassium	1.67	1.73	1.42	1.55	1.19
Sodium	0.25	0.30	0.27	0.27	0.45

^1^ Composition not less than: 95.5% NaCl, 0.24% Mn, 0.24% Fe, 0.05% Mg, 0.032% Cu, 0.032% Zn, 0.007% I, and 0.004% Co. ^2^ Contributed 4400 IU Vitamin A; 2200 IU Vitamin D; and 16 IU Vitamin E per kg of dietary DM. ^3^ Provided 0.06 mg Se per kg. ^4^ Dry matter.

**Table 4 animals-11-01230-t004:** Prepartum and postpartum body weight (BW) and body condition scored (BCS) of primiparous and multiparous cows fed control or molasses-containing diets during a 60 day dry period (Experiment 1).

Item/Day to Parturition	Control		Molasses		SEM	*p*Value		
	Primiparous	Multiparous	Primiparous	Multiparous		Diet	Parity	Diet × Parity
BW, kg								
−60	655	704	662	717	12.5	0.55	0.004	0.86
−30	676	732	687	735	10.4	0.62	<0.001	0.79
0	631	690	645	717	10.8	0.19	<0.001	0.65
30	549	615	576	637	10.0	0.08	<0.001	0.89
75	555	630	584	653	11.5	0.10	<0.001	0.85
BCS ^1^								
−60	3.67	2.79	3.82	2.92	0.057	0.08	<0.001	0.89
−30	3.71	2.96	3.78	3.03	0.054	0.33	<0.001	0.99
0	3.71	3.08	3.75	3.05	0.050	0.93	<0.001	0.64
30	2.83	2.42	2.91	2.52	0.068	0.35	<0.001	0.96
75	2.73	2.22	2.80	2.38	0.069	0.20	<0.001	0.68

^1^ Scored on 1 to 5 scale [16]. SEM: Standard error of the mean.

**Table 5 animals-11-01230-t005:** Prepartum and postpartum BW change and BCS change for primiparous and multiparous cows fed control or molasses diets during a 60 day dry period (Experiment 1).

Item/Day to Parturition	Control		Molasses		SEM	*p*Value		
	Primiparous	Multiparous	Primiparous	Multiparous		Diet	Parity	Diet × Parity
Body weight change, kg								
−60 to −30	21.2	27.4	24.8	17.3	5.11	0.65	0.93	0.33
−30 to 0	35.2	38.6	37.5	62.0	6.79	0.18	0.14	0.26
0 to 30	−82.3	−75.9	−69.0	−79.8	8.16	0.68	0.85	0.45
0 to 75	−76.0	−77.3	−60.2	−62.7	12.10	0.34	0.91	0.97
Body condition change^1^								
−60 to −30	0.04	0.17	−0.04	0.17	0.455	0.30	0.03	0.88
−30 to 0	0.00	0.12	−0.04	0.02	0.035	0.17	0.09	0.52
0 to 30	−0.88	−0.65	−0.84	−0.53	0.059	0.31	0.001	0.64
0 to 75	−1.00	−0.86	−0.95	−0.66	0.077	0.21	0.04	0.47

^1^ Scored on 1 to 5 scale [16]. SEM: Standard error of the mean.

**Table 6 animals-11-01230-t006:** Plasma urea nitrogen (PUN), glucose, and non-esterified fatty acids (NEFA) concentrations from blood collected on day 1, 3, 5, 7, and 15 of lactation for primiparous and multiparous cows fed control or molasses-containing diets during a 60 day dry period (Experiment 1).

Item	Control		Molasses		SEM	*p*Value		
	Primiparous	Multiparous	Primiparous	Multiparous		Diet	Parity	Diet × Parity
PUN, mg/dL	11.7	13.8	11.7	14.3	0.42	0.52	<0.0001	0.54
Glucose, mg/dL	62.0	59.3	64.0	59.7	2.10	0.55	0.09	0.69
NEFA, µmol/L	732	532	620	523	47.7	0.20	0.002	0.28

SEM: Standard error of the mean.

**Table 7 animals-11-01230-t007:** Dry matter intake (DMI) and milk yield and composition during the initial 60 day of lactation for cows fed control or molasses-containing diets during a 60 day dry period (Experiment 2).

Item	Diet		SEM	*p*-Value
	Control	Molasses		
DMI, kg/day				
Far-off	13.0	13.1	0.97	0.96
Close-up	12.1	13.6	1.19	0.002
Lactation	23.3	26.5	2.6	0.08
Milk yield, kg/day	40.4	44.3	4.1	0.41
ECM ^1^, kg/day	38.7	45.9	4.9	0.07
Milk fat				
%	3.57	3.62	0.18	0.83
kg/day	1.33	1.68	0.22	0.002
Milk protein				
%	2.89	3.02	0.087	0.06
kg/day	1.17	1.32	0.092	0.07
Milk lactose, %	4.84	3.02	0.087	0.06
SCC ^2^, per µL	25	32	5.4	0.22
Milk urea nitrogen, mg/dL	14.3	17.4	1.7	0.001

^1^ Energy corrected milk; ^2^ Somatic cell count; SEM: Standard error of the mean.

**Table 8 animals-11-01230-t008:** Prepartum and postpartum Body weight (BW) and BCS (Body Condition Score) of cows fed control or molasses-containing diets during a 60 day dry period (Experiment 2).

Item/Day to Parturition	Diet		SEM	*p*-Value
	Control	Molasses		
BW, kg				
−60	675	651	15	0.32
−30	699	680	10	0.27
−2	737	716	23	0.56
0	697	671	24	0.50
15	687	653	24	0.38
30	681	646	15	0.17
60	680	656	20	0.49
BCS ^1^				
−60	2.67	2.58	0.08	0.52
−30	2.75	2.67	0.16	0.73
−2	2.92	3.00	0.12	0.64
0	3.08	3.17	0.08	0.52
15	2.73	2.75	0.01	0.38
30	2.57	2.58	0.13	0.95
60	2.67	2.50	0.13	0.37

^1^ Scored on 1 to 5 scale [16]. SEM: Standard error of the mean.

**Table 9 animals-11-01230-t009:** Body weight change and body condition change for cows fed control or molasses diets during a 60 day dry period (Experiment 2).

	Diet		SEM	*p* Value
	Control	Molasses		
Body weight change, kg				
−60 to −30	24.6	29.6	9.0	0.71
−30 to −2	37.9	35.6	15.3	0.92
0 to 15	−9.8	−18.2	7.0	0.45
15 to 30	−6.1	−7.6	18.4	0.96
30 to 60	−1.5	12.1	11.7	0.46
Body condition change ^1^				
−60 to −30	0.08	0.08	0.08	1.00
−30 to −2	0.17	0.33	0.08	0.23
0 to 15	−0.35	−0.43	0.08	0.59
15 to 30	−0.16	0.17	0.13	0.99
30 to 60	0.09	−0.10	0.08	0.15

^1^ Scored on 1 to 5 scale [16]. SEM: Standard error of the mean.

**Table 10 animals-11-01230-t010:** Diet organic matter (OM), neutral detergent fiber (NDF), and acid detergent fiber (ADF) digestibility of far-off and close-up diets by cows fed either control or molasses-containing diets during the dry period (Experiment 2).

	Far-off		Close-up		SEM	*p*Value		
Item	Control	Molasses	Control	Molasses		Diet	Period	Diet × Period
OM digestibility, % intake	51.6	55.6	61.5	56.1	2.6	0.75	0.05	0.10
NDF digestibility, % intake	42.7	47.5	49.4	46.3	2.3	0.79	0.40	0.24
ADF digestibility, % intake	34.5	37.9	44.3	39.4	2.9	0.85	0.19	0.32

SEM: Standard error of the mean.

**Table 11 animals-11-01230-t011:** Rumen fluid volatile fatty acids (VFA) proportions, pH, and kinetics during the prepartum period for cows fed either control or molasses-containing diets during a 60-day dry period (Experiment 2).

Item	Diet		SEM	Period		SEM	*p*Value		
	Control	Molasses		Far-off	Close-up		Diet	Period	Diet × Period
Total VFA, mM	97	88	7.89	87	98	7.72	0.31	0.02	0.53
Acetate, molar %	68.9	67.1	1.36	69.2	66.9	1.17	0.39	0.04	0.36
Propionate, molar %	16.9	18.5	1.15	17.0	18.5	0.95	0.41	0.003	0.95
Butyrate, molar %	11.7	12.3	0.44	11.7	12.2	0.42	0.37	0.42	0.33
Isovalerate, molar %	1.36	1.21	0.11	1.22	1.35	0.12	0.27	0.33	0.68
Rumen parameter									
pH	7.11	7.16	0.19	7.54	7.09	0.16	0.62	0.04	0.73
Liquid dilution, %/h	7.9	12.0	2.32	7.3	12.6	2.48	0.32	0.15	0.61
Valerate absorption, %/h	31.2	32.8	4.43	25.0	39.0	3.55	0.79	0.02	0.69
Rumen liquid volume, L	57.6	51.5	5.63	58.3	50.9	6.17	0.47	0.39	0.85
Outflow, L/h	4.7	5.8	1.19	4.2	6.3	1.17	0.56	0.21	0.77
Turnover, h	12.9	10.8	2.61	14.6	9.0	2.87	0.53	0.14	0.50

SEM: Standard error of the mean.

**Table 12 animals-11-01230-t012:** Rumen fluid volatile fatty acids proportions, pH, and kinetics during the postpartum period for cows fed either a control or a molasses-containing diet during a 60 day dry period (Experiment 2).

Item	Prepartum Diet		SEM	Day Postpartum						SEM	*p*Value		
	Control	Molasses		2	16	30	44	58	72		Diet	Day	Diet × Day
Total VFA, mM	101.8	101.9	4.33	108.6	97.0	100.9	100.0	100.6	103.9	5.00	0.98	0.20	0.13
Acetate, molar %	58.1	58.1	1.64	58.8	55.5	58.7	58.5	58.6	58.3	1.85	0.99	0.40	0.28
Propionate, molar %	23.4	25.8	0.93	22.8	27.4	24.4	24.8	24.8	23.4	1.16	0.11	0.02	0.16
Butyrate, molar %	15.5	13.9	0.86	16.5	15.0	14.6	14.4	13.7	14.0	0.83	0.21	0.12	0.23
Isovalerate, molar %	1.78	1.34	0.31	1.14	1.32	1.45	1.45	1.65	2.33	0.45	0.38	0.43	0.62
Rumen parameter													
pH	6.87	6.76	0.24	6.99	6.73	6.61	6.75	6.72	6.81	0.19	0.63	0.58	0.74
Liquid dilution, %/h	14.7	17.2	0.92	12.8	17.0	17.4	17.7	14.9	16.3	1.38	0.11	0.03	0.37
Valerate absorption, %/h	35.3	43.2	4.53	33.2	50.5	39.2	30.2	43.3	39.0	7.63	0.21	0.48	0.84
Liquid volume, L	68.5	58.7	4.08	62.5	60.6	66.1	68.7	62.9	60.9	4.91	0.20	0.72	0.48
Outflow, L/h	9.9	10.1	1.05	7.9	10.1	11.4	13.1	9.2	9.6	1.08	0.93	0.002	0.19
Turnover, h	9.4	7.4	0.80	9.9	8.1	7.4	7.8	8.8	8.2	0.88	0.17	0.17	0.22

SEM: Standard error of the mean.

## Data Availability

The data presented in this study are available on request from the corresponding author.

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
