# Peer review of "Influence of Cane Molasses Inclusion to Dairy Cow Diets during the Transition Period on Rumen Epithelial Development"

_animals, 2021, doi:10.3390/ani11051230_

Round 1
Reviewer 1 Report
The manuscript derives from a very complex and demanding work.
Simple summary and Abstract are not so clear to allow the reader to
understand the article content.
Some explanations of abbreviations are missing the first time they appear.
There is a serious problem with rumen morphometric evaluation.
Authors stated that after biopsy, rumen papillae were fixed;
it is not clear if morphometry was carried out on fixed or fresh papillae.
Fixation can slithly alter the size of structure.
In additon, the rumen absorpite ability have to be assessed by
Surface Enlargement Factor evaluation; Authors didn't extimated the papillae density.
To assess a comparison of palillae between two different diets or times or areas and so on,
it is not sufficient to measure papillae height and width,
but the exact area of papillae surface.
I suggest to read the article that could aid Authors to
better perform their evaluation
DOI 10.1002/jemt.22692
Author Response
Reviewer 1:
The manuscript derives from a very complex and demanding work.
Simple summary and Abstract are not so clear to allow the reader to
understand the article content.
Some explanations of abbreviations are missing the first time they appear.
There is a serious problem with rumen morphometric evaluation.
Authors stated that after biopsy, rumen papillae were fixed;
it is not clear if morphometry was carried out on fixed or fresh papillae.
Fixation can slithly alter the size of structure.
In additon, the rumen absorpite ability have to be assessed by
Surface Enlargement Factor evaluation; Authors didn't extimated the papillae density.
To assess a comparison of palillae between two different diets or times or areas and so on,
it is not sufficient to measure papillae height and width,
but the exact area of papillae surface.
I suggest to read the article that could aid Authors to
better perform their evaluation
DOI 10.1002/jemt.22692
We would like to thank this reviewer for bringing to our attention the article describing a method on rumen surface enlargement factor. However, the article suggested by this reviewer used slaughtered sheep to perform this method, which was not the case of our study that used cannulated cows. In addition, in the article suggested by this reviewer, authors collected a 5 x 5 cm fragment from each rumen of sheep that had been slaughtered, which would not be feasible for live animals; since the size of the fragment collected would have to be adjusted for large ruminants, in our case: cows. Therefore, conducting biopsies to sample large fragments from the rumen epithelium of live cannulated cows is really invasive and may be detrimental to rumen function and animal health. If we had slaughtered our cows, certainly it would be the method of our choice; however, we decided to use a less invasive method to determine rumen absorptive ability by using CoEDTA and valeric acid, and combining it with rumen morphometry measures that are reported in this study. We may agree with this reviewer that sometimes fixing may slightly change the size of the papillae; however, since we did not measure papillae density per square cm2 of rumen wall, and as result, we did not used these measurements to calculate the rumen absorptive surface area, all samples were fixed with hematoxylin and eosin. Finally, our group has published extensive work on papillae evaluations when sampling takes place in either the ventral or cranial sac, which is the choice when we need to determine the rumen absorptive surface area in slaughtered animals (10.1111/jpn.13542; 10.1071/an18657; 10.1017/S1751731120001147; 10.1016/j.livsci.2020.103985; 10.1017/ S002185962000026X; 10.3168/jds.2020-18514).
Reviewer 2 Report
thanks for having the opportunity to review this work. the article is very long, but well written and organized. Below are my comments.
Line 137 and 138 and in results: may be better “lipid” instead “fat”
Line 140: “by” instead “By”
Line 154: could you show the statistical model of your analysis?
Line 159-160: you should use two level of statistical significance: p < 0.05 and p < 0.01
Line 170: I did not understand why do you use two different diet and not a same diet into the 2 experiment
Figures: sincerely I don’t like this type of graphic.
Author Response
Reviewer 2:
Thanks for having the opportunity to review this work. the article is very long, but well written and organized. Below are my comments.
Thank you. Comments addressed below and changes were made to the manuscript in red.
Line 137 and 138 and in results: may be better “lipid” instead “fat”
We prefer fat since individual lipid fractions were not determined and crude fat was determined.
Line 140: “by” instead “By”
Corrected on L 140
Line 154: could you show the statistical model of your analysis?
Details are listed on L 154 to 162
Line 159-160: you should use two level of statistical significance: p < 0.05 and p < 0.01
Done L 161
Line 170: I did not understand why do you use two different diet and not a same diet into the 2 experiment
This was a product of the research farm and times of cows fed.
Figures: sincerely I don’t like this type of graphic.
Not sure what to include aside form a figure such as this.
Reviewer 3 Report
This is a very nice, well written article which offers some fantastic insights into the alteration of dairy cow diets using molasses. It uses a wide range of techniques which are highly appropriate and shows some fantastic results which will be useful to the wider field. I wish to offer my congratulations to the authors on such a well executed and well written study, and offer my best wishes for their future research.
I have a few very minor comments below. But feel free to ignore any you don’t agree with
One thing I struggled a bit with (and may just be my covid infected brain not quite at full speed) but there are a lot of abbreviations in here, and I kept having to look back to check what each one meant. Would an abbreviations section be useful? May depend on the journal requirements too
Line 17- I think a word is missing ‘a lactating diet can a substantial amount of time’?
Line 46-48- I struggled to follow this (but that may just be me being a bit thick). Possibly consider rewording?
Line 52- Moreover, the same authors (reword)
Line 64- This is well worded however is a little difficult to follow. Would a figure here be useful? Entirely up to the authors
Line 103- you define TMR in line 122, but not here, could you move those around please?
Table 2- This may just be formatting, but is this not results?
Line 122- you say that feed was offered twice daily and the amount refused measured. Was this left all day, or only there for a certain time? A bit of clarification there may help
Line 140- I would be tempted to have the authors name in here or it doesn’t read very well for reference 17. Same true for line 149 and reference 20, and line 571 and reference 30
Line 167- comma needed after design
Line 195- I may have missed it, but whats ADIA ?
Line 216- Maybe having Co as cobalt the first time may make it clearer?
Line 261- maybe worth defining PBST?
Section 3.1.1. This is very nice, but I wonder if some sort of figure or table may help here just for clarity? Also, including the actual data here may be nice as well?
Line 329- comma after expected
Line 332- with the control diet (reword)
Line 365- I don’t think that ‘for in this experiment’ make sense? Consider rewording
Line 366- fed control diets (reword)
Line 410- Again this may be me being a bit thick- but NDF and ADF appear to be the same thing?
Line 420- diet man effect sounds a bit strange? Again maybe reword. And in line 431
Line 420- any of the rumen parameters (reword)
Line 448- cows fed the control diet (reword)
Line 489-493- is this not methods? (more a question than suggesting that it needs moving)
Line 531- I think I know what you are saying here but it is a little unclear. Could you reword please? I read it a few times as ‘day tended to differ’ and wondered if time stood still for the other group!?
Line 552- maybe resulting in improved lactation may sound better?
Line 606-610- I struggled to follow this a bit, please consider rewording
Line 616- maybe ‘those fed the control diet’ maybe easier to follow?
Line 618- offset the effects (reword)
Line 640- The results and discussion read very well. I was wondering if it was possible to include some comment on cost increases vs benefits for the farmer to ascertain very clearly if this would be worth considering for animals ?
Author Response
Reviewer 3:
This is a very nice, well written article which offers some fantastic insights into the alteration of dairy cow diets using molasses. It uses a wide range of techniques which are highly appropriate and shows some fantastic results which will be useful to the wider field. I wish to offer my congratulations to the authors on such a well executed and well written study, and offer my best wishes for their future research.
Thank you. Comments addressed below and changes were made to the manuscript in red.
I have a few very minor comments below. But feel free to ignore any you don’t agree with
One thing I struggled a bit with (and may just be my covid infected brain not quite at full speed) but there are a lot of abbreviations in here, and I kept having to look back to check what each one meant. Would an abbreviations section be useful? May depend on the journal requirements too
Agreed, there are many abbreviations and this “list” is the norm. in some journals, however, all abbreviations are defined at first use.
Line 17- I think a word is missing ‘a lactating diet can a substantial amount of time’?
Corrected L 17.
Line 46-48- I struggled to follow this (but that may just be me being a bit thick). Possibly consider rewording?
Corrected L 51 to 52.
Line 52- Moreover, the same authors (reword)
Corrected L 52.
Line 64- This is well worded however is a little difficult to follow. Would a figure here be useful? Entirely up to the authors
Prefer to leave as is.
Line 103- you define TMR in line 122, but not here, could you move those around please?
Corrected L 103.
Table 2- This may just be formatting, but is this not results?
Agreed and this should be moved during final type setting if accepted for publication..
Line 122- you say that feed was offered twice daily and the amount refused measured. Was this left all day, or only there for a certain time? A bit of clarification there may help
Corrected L 123 to 124.
Line 140- I would be tempted to have the authors name in here or it doesn’t read very well for reference 17. Same true for line 149 and reference 20, and line 571 and reference 30
This is MDPI citation format.
Line 167- comma needed after design
Corrected L 161.
Line 195- I may have missed it, but whats ADIA ?
L 195 acid detergent insoluble ash (ADIA)
Line 216- Maybe having Co as cobalt the first time may make it clearer?
Corrected on L 218
Line 261- maybe worth defining PBST?
Corrected on L 263
Section 3.1.1. This is very nice, but I wonder if some sort of figure or table may help here just for clarity? Also, including the actual data here may be nice as well?
All data are in the M.S.
Line 329- comma after expected
Corrected L 331
Line 332- with the control diet (reword)
See L 334
Line 365- I don’t think that ‘for in this experiment’ make sense? Consider rewording
Removed ‘ for ‘
Line 366- fed control diets (reword)
Corrected L 368
Line 410- Again this may be me being a bit thick- but NDF and ADF appear to be the same thing?
Corrected on L 412
Line 420- diet man effect sounds a bit strange? Again maybe reword. And in line 431
See L 433.
Line 420- any of the rumen parameters (reword)
See L 433
Line 448- cows fed the control diet (reword)
See L 450
Line 489-493- is this not methods? (more a question than suggesting that it needs moving)
This was to direct the reader to know we did things appropriately.
Line 531- I think I know what you are saying here but it is a little unclear. Could you reword please? I read it a few times as ‘day tended to differ’ and wondered if time stood still for the other group!?
Excellent point. Corrected on L
Line 552- maybe resulting in improved lactation may sound better?
Corrected on L 533
Line 606-610- I struggled to follow this a bit, please consider rewording
Modified on L 608 to 611
Line 616- maybe ‘those fed the control diet’ maybe easier to follow?
Modified on L 618
Line 618- offset the effects (reword)
Modified on L 620
Line 640- The results and discussion read very well. I was wondering if it was possible to include some comment on cost increases vs benefits for the farmer to ascertain very clearly if this would be worth considering for animals ?
Economic analyses are not static. We wish to provide underlying data for a more robust economic assessment conducted in the future using meta-analysis procedures.
Round 2
Reviewer 1 Report
Authors gave some explainations about their choice as morphometrics regards.
In their response Authors stated that the samples were fixed with hematoxylin and eosin, but this is a morphological staining and not a fixative medium; however in the text the sentence is correct. This fact introduces some doubts about the Authors accuracy.
However, in my opinion, some of their evaluation, also in previously published articles, may be non so fine.
The rumen absorpive ability depends on both SEF and epithelium keratinization degree that cannot be expressed by keratinazed layer height, but have to be considered as its ratio respect to the total epithelial height.
Authors have at least to briefly discuss their choice in comparison with the suggested articles:
DOI 10.1002/jemt.22008
DOI 10.1002/jemt.22692
Author Response
We agree with this reviewer that rumen absorptive ability depends on epithelium keratinization and SEF, but unfortunately, we did not measure any of these variables in this study. Moreover, based on the references provided by this reviewer, we decided to change what we defined as “rumen absorptive ability” to simply “valerate absorption” (as already shown on Tables 11 and 12), which is what we really measured. We hope to have addressed all concerns of this reviewer sufficiently with respect to rumen parameters.
These changes have been made to manuscript in the yellow highlighted sections.